# Do Rare Genetic Conditions Exhibit a Specific Phonotype? A Comprehensive Description of the Vocal Traits Associated with Crisponi/Cold-Induced Sweating Syndrome Type 1

**DOI:** 10.3390/genes16080881

**Published:** 2025-07-26

**Authors:** Federico Calà, Elisabetta Sforza, Lucia D’Alatri, Lorenzo Frassineti, Claudia Manfredi, Roberta Onesimo, Donato Rigante, Marika Pane, Serenella Servidei, Guido Primiano, Giangiorgio Crisponi, Laura Crisponi, Chiara Leoni, Antonio Lanatà, Giuseppe Zampino

**Affiliations:** 1Department of Information Engineering, University of Florence, 50139 Florence, Italy; federico.cala@unifi.it (F.C.); lorenzo.frassineti@unifi.it (L.F.); claudia.manfredi@unifi.it (C.M.); antonio.lanata@unifi.it (A.L.); 2Faculty of Medicine and Surgery, Università Cattolica del Sacro Cuore, 00168 Rome, Italy; elisabetta.sforza@unicatt.it; 3Unit for Ear, Nose and Throat Medicine, Department of Neuroscience, Sensory Organs and Chest, Fondazione Policlinico Universitario Agostino Gemelli IRCCS, 00168 Rome, Italy; lucia.dalatri@policlinicogemelli.it; 4Department of Woman and Child Health and Public Health, Fondazione Policlinico Universitario Agostino Gemelli IRCCS, 00168 Rome, Italy; roberta.onesimo@policlinicogemelli.it (R.O.); donato.rigante@unicatt.it (D.R.); chiara.leoni@policlinicogemelli.it (C.L.);; 5Centro Clinico Nemo Pediatrico, Fondazione Policlinico Universitario Agostino Gemelli IRCCS, 00168 Rome, Italy; marika.pane@policlinicogemelli.it; 6Dipartimento di Neuroscienze, Organi di Senso e Torace, Fondazione Policlinico Universitario Agostino Gemelli IRCCS, 00168 Rome, Italy; serenella.servidei@unicatt.it (S.S.); guidoalessandro.primiano@policlinicogemelli.it (G.P.); 7Independent Researcher, 09042 Cagliari, Italy; crisponi@gmail.com; 8Institute for Genetic and Biomedical Research (IRGB), The National Research Council (CNR), Monserrato, 09042 Cagliari, Italy; laura.crisponi@cnr.it

**Keywords:** Crisponi/cold-induced sweating syndrome, genetics, machine learning, phonotype, human phenotype ontology

## Abstract

**Background**: Perceptual analysis has highlighted that the voice characteristics of patients with rare congenital genetic syndromes differ from those of normophonic subjects. In this paper, we describe the voice phenotype, also called the phonotype, of patients with Crisponi/cold-induced sweating syndrome type 1 (CS/CISS1). **Methods**: We conducted an observational study at the Department of Life Sciences and Public Health, Rome. Thirteen patients were included in this study (five males; mean age: 16 years; SD: 10.63 years; median age: 12 years; age range: 6–44 years), and five were adults (38%). We prospectively recorded and analyzed acoustical features of three corner vowels [a], [i], and [u]. For perceptual analysis, the GIRBAS (grade, instability, roughness, breathiness, asthenia, and strain) scale was utilized. Acoustic analysis was performed through BioVoice software. **Results**: We found that CS/CISS1 patients share a common phonotype characterized by articulation disorders and hyper-rhinophonia. **Conclusions**: This study contributes to delineating the voice of CS/CISS1 syndrome. The phonotype can represent one of the earliest indicators for detecting rare congenital conditions, enabling specialists to reduce diagnosis time and better define a spectrum of rare and ultra-rare diseases.

## 1. Introduction

Crisponi/cold-induced sweating syndrome type 1 (CS/CISS1, OMIM #272430) is an autosomal recessive congenital disorder, first described three decades ago by Giangiorgio Crisponi [1,2]. Although this genetic condition is ultra-rare, a higher prevalence is recorded in Sardinia (Italy) with an incidence of 1 case in every 20,700 newborns [3,4]. The differential diagnosis is placed between two other conditions with similar phenotypes: cold-induced sweating syndrome type 2 (CISS2, OMIM #610313) and Stüve–Wiedemann syndrome/Schwartz-Jampel type 2 syndrome (SWS/SJS2, OMIM #601559) [4]. CS/CISS1 is caused by variants in the cytokine receptor-like factor 1 (*CRLF1*) gene, coding for a ligand of the ciliary neurotrophic factor receptor (CNTFR) [2].

Its phenotype is characterized by a tendency towards hyperthermia, cold-induced sweating, camptodactyly, kyphoscoliosis, and respiratory and/or feeding difficulties [1]. The striking facial features of CS/CISS1 include chubby cheeks, a large face, puckered lips, broad nose with anteverted nares, and a long philtrum [1]. Feeding difficulties are constantly present at birth, caused by facial muscle contractions and orofacial weakness [5]. The muscles involved are the mouth and masseter muscles, causing a limited mouth opening. Moreover, the presence of trismus observed in CS/CISS1 is commonly associated with signs and symptoms of trigeminal sensory and palatine veil dysfunction, causing nasal regurgitation [6].

Among the striking features of this condition, voice characteristics have not yet been defined. Moreover, given the low prevalence, patients’ data are scarce and scattered, with only single cases reported in the medical literature [7]. Since voice assessment is based on non-invasive and easily administered tests, vocal characterization could represent an informative tool in the diagnostic process of rare under-recognized conditions and help define the severity of clinical pictures. Consequently, one of the main challenges in the field of genetic diseases is collecting data from large study cohorts. In the present study, we gathered consistent data from a relatively large cohort of Italian CS/CISS1 patients and analyzed it to enhance our understanding of voice phenotype, also referred to as phonotype.

## 2. Participants and Methods

### 2.1. Participants

This research starts with observations of cohorts of rare conditions recruited through the “AD MAIORA” project (PNRR-MR1-2022-12376346). For the purpose of this study, patients with a confirmed molecular diagnosis of CS/CISS1 syndrome were prospectively recruited over a six-month period at the Department of Life Sciences and Public Health, Fondazione Policlinico Agostino Gemelli-IRCCS, Rome, Italy. All these patients were previously reported [3,4,5,6,8]. No age restrictions were set. The patients were included in the study after signed informed consents was secured.

For each patient, the clinical, laboratory and instrumental data collected were entered into an eCRF, according to MONDO Disease Ontology coding, human phenotype ontology (HPO), and Anatomical Therapeutic Chemical (ATC) coding. The data for each enrolled patient were discussed by the Multidisciplinary Board for Rare Diseases (MBRD), composed of pediatricians, geneticists, child neurologists, biologists, psychologists and case managers.

Thirteen patients were enrolled in this study. They were divided into three subgroups according to age and gender: 6 pediatric subjects (PS, patients with age ≤ 13, mean age = 8.3 ± 2.0 years), 6 adult females (AF, mean age = 24.2 ± 11.6 years) and 1 adult male (AM, age = 25). None of the participants included were siblings to other participants.

All patients were of Italian ancestry, and in most cases, they were Sardinian (*n* = 10/13, 77%). The Local Ethical Committee approved the study (approval n.5518 of 2023), and all research procedures were in line with the Declaration of Helsinki. All subjects carried a *CRLF1* mutation, either in a homozygous or compound heterozygous state. Among the clinical features, chubby cheeks and micrognathia were found in thirteen (100%) and seven patients (53%), respectively, while high arched palate and anteverted nostrils were found in nine (69%) and eleven (84%) patients. An expressionless face was observed in seven patients (53%), while trismus was observed in eleven patients (84%).

The age- and gender-matched healthy control (HS) group consisted of 45 non-smoking subjects. In the 3–13 years subgroup, a total of *n* = 20 children were included (mean age 8.5 years ± 2.3; median age 9 years; age range 5–13 years); in the 13 years and older subgroup, a total of *n* = 25 adults were included, comprising *n* = 10 adult females (mean age 28.2 years ± 5.7; median age 29.5; age range 16–38 years) and *n* = 15 adult males (mean age 28 years ± 5.9; median age 30 years; age range 15–35 years).

### 2.2. Vocal Task

The voice recording protocol required each participant to produce a message containing the three Italian corner vowels /a/, /i/, and /u/, each prolonged for at least 3 s at conversational pitch and loudness. The recordings were made in a silent environment (<40 dB of background noise) using the integrated microphone of a smartphone [9], which was placed at a constant distance of about 15 cm from the lips (to avoid interference from ambient noise) and angled at 45° (to avoid the disruption of airflow) [10]. Manual segmentation of individual vowel utterances was performed in Audacity (Audacity Team 2011); additionally, vocal attack and extinction were excluded from the analysis to omit the most unstable portions of vocalization. The choice of the vocal task, as well as the feature extraction pipeline explained in Section 2.4, followed the recommendations introduced in [10]. The latter represents an adapted version of the SIFEL (Società Italiana di Foniatria e Logopedia) protocol which was designed to be used with patients with genetic syndromes, as it accounts for their voice production and behavioural impairments.

### 2.3. Perceptual Analysis

For perceptual analysis, the GIRBAS (grade, instability, roughness, breathiness, asthenia, and strain) scale was completed by one otorhinolaryngologist (LD) and one speech language pathologist (ES) considering the entire audio file containing the patient’s vowel production. Each dimension was rated on a 4-point Likert scale where 0 refers to no perceived abnormality and 3 corresponds to severe dysphonia (human phenotype ontology-HPO identifier: 0001618) [11,12,13].

Speech and resonance disorders were also noted [14]. Specifically, the presence of speech articulation difficulties (HPO identifier: 0009088) and hypernasality (HPO identifier: 0001611) were recorded [15] in order to standardize the description of voice phonotype.

### 2.4. Objective Acoustical Analysis

Acoustic analysis was conducted using the BioVoice software tool (version 1.0) [16], which automatically determines the appropriate frequency ranges with which to extract acoustic parameters, covering both frequency and time domains. Indeed, the calculations are based on the type of vocal emission (voice, crying, or singing), the individual’s age (child or adult), and their gender (male or female), which the users can manually set.

The characterization of the three sustained vowel emissions included the fundamental frequency (F0, Hz) and its standard deviation, the local jitter (in %), normalized noise energy (NNE, dB), and the first three formants (F1, F2, and F3, Hz), along with their respective standard deviations. These latter parameters primarily reflect the influence of the supraglottic filter rather than the laryngeal oscillator and may highlight potential articulation alterations.

Additionally, five key metrics were derived: the vowel space area (VSA), formant centralization ratio (FCR), and three formant ratios. VSA serves as a measure of vowel dispersion, linked to articulatory capability [17], while FCR, introduced by Sapir et al. [18], is used to evaluate dysarthria severity (a higher FCR value indicates a potentially pathological vowel centralization). The formant ratios help track formant trajectories and assess articulatory proficiency, with the F1a/F1i and F1a/F1u ratios being sensitive to vertical tongue movements and the F2i/F2u ratio being sensitive to horizontal ones [18]. In total, 35 parameters were computed, of which 10 features refer to each corner vowel, and 5 features refer to derived formant metrics.

### 2.5. Statistical Analysis

Despite the small sample size, statistical analyses were performed by comparing the group diagnosed with the syndrome and the control, taking into account age and gender, in order to highlight vocal features unique to specific population subgroups. After checking for data normality using a Shapiro–Wilk test, a Mann–Whitney *U* test with a significance level of α set at 0.05 was conducted for each objective phonation and articulation parameter introduced in Section 2.4, since perceptual evaluation was not carried out for the control group. The strength of agreement among perceptual analysis assessors (inter-rater reliability) was calculated by Krippendorff’s α.

## 3. Results

The results from the GIRBAS evaluation scale are presented in Table 1 as the median (IQR), where IQR stands for the interquartile range. The level of agreement between the evaluators was high (α = 0.8). Moreover, perceptual evaluation also reported the presence of speech articulation difficulties (*n* = 10, 83%,) and hypernasality (*n* = 8, 66%), as shown in Table 2. Notably, the value of 1 represents a slight alteration from normophonic vocal properties, whereas 0 indicates no alteration.

Only one adult female patient exhibited a severe grade of dysphonia characterized by a higher level (3) of breathiness and asthenia due to unilateral vocal cord paralysis.

Concerning acoustic analysis, Appendix A reports the means and standard deviations for the extracted parameters from each corner vowel and both healthy and pathological populations. Data are further divided by age and gender, where the acronym PS refers to pediatric subjects (age range 6–12 years), AF refers to adult females, AM refers to adult males, and HS refers to healthy subjects.

Statistically significant differences in the PS group were found for the F1 mean /a/ (*p* = 0.02), normalized noise energy (NNE) /i/ (*p* = 0.01), F0 std /u/ (*p* = 0.02), NNE /u/ (*p* = 0.03), F1a/F1i ratio (*p* = 0.04), F1a/F1u ratio (*p* = 0.04), and the VSA (*p* < 0.001). The boxplots in Figure 1 show that pediatric Crisponi (CS/CISS1 patients) voices are characterized by noisier voices especially when uttering vowels /i/ and /u/, as well as a higher F0 std for /u/. Concerning articulation capabilities, F1 mean /a/ is significantly lower than age-matched healthy controls and the formant ratios F1a/F1i and F1a/F1u, along with the VSA, are also smaller.

The AF group presented similar results, as both NNE /i/ and NNE /u/ were found to be significantly closer to 0 than to the HS population (*p* = 0.004 and *p* = 0.03, respectively). Moreover, significant differences were discovered for articulation parameters, specifically for the F1 std /i/ (*p* = 0.02) and F3 mean /i/ (*p* = 0.02), as shown in the boxplots in Figure 2.

## 4. Discussion

Voice analysis is crucial in diagnosing syndromic disorders and might serve as a phenotypic marker. Numerous genetic syndromes exhibit distinct voice anomalies that assist in the early detection and categorization of these disorders. Genetics influences voice features since it is responsible for physiology and the structures involved in speech production [19,20]. Alterations in voice production can result from factors like vocal cord function, respiratory support, neuromuscular control, or hormonal influences. In terms of voice analysis methodology, the present study follows the procedure used to evaluate the phono types of Smith–Magenis, Costello, Noonan, Down, and Cri du Chat syndromes [14,15,16]. In the present study, we found that CS/CISS1 patients share a common phonotype characterized by not only the hypernasality already reported within the HPO for this condition (identifier: 0001611), but also by the presence of speech articulation difficulties (HPO identifier: 0009088). Specifically, the HPO provides a standardized vocabulary, that is crucial for clinical and research applications such as the following:

Diagnostic Orientation—Voice characteristics can serve as early indicators of syndromic disorders, guiding genetic testing and differential diagnosis.

Phonotype-Genotype Correlation—standardized voice descriptors in HPO can facilitate research into genetic mutations associated with specific voice anomalies. Furthermore, HPO enables the production of standardized clinical documentation—ensuring consistency in describing voice-related symptoms across studies.

Therapeutic Planning—Understanding voice impairments in syndromic disorders allows for targeted speech therapy and intervention strategies.

This was highlighted through perceptual analysis, which is already considered a gold standard for documenting voice disorders [14,15]. The acoustic analysis showed that the phonatory activity of Crisponi patients does not significantly deviate from that of healthy controls. Indeed, the mean values of the fundamental frequency F0 for each corner vowel in the PS and AF groups are comparable with those of age- and gender-matched healthy populations. In contrast, the values from the AM subject are approximately 30 Hz higher than those of the corresponding HS group, which may indicate a potentially higher pitch among male patients. However, more data is necessary to validate such a hypothesis. Notably, the perceptual evaluation did not reveal any alteration in pitch regarding the pathological population. On the other hand, statistical analysis revealed that the NNE of /i/ and /u/ is closer to 0 compared to control subjects in the pediatric subgroup, as both Table 2 and Figure 1 demonstrate. A higher noise in voice production could be caused by laryngospasms and neck muscles hypertonia, which are common phenotypical characteristics of patients diagnosed with this syndrome [4], or hyper nasal speech, which is a quality noted by perceptual assessment. Moreover, hyper nasal speech can also be linked to a trigeminal motor dysfunction, causing the inhibition of the palatine veil with nasal regurgitation and an excessive passage of air from the nose during phonation [6].

Interestingly, the medians of the roughness and breathiness indices, which usually correlated with vocal noise measures [21,22], are 0: This could be explained by the fact that the NNE remains below −20 dB (as Table 2 reports) and is a property that is usually associated with normophonic voices [23]. Other medical signs of the CS/CISS1 syndrome include limited jaw movements and palatal deformations, which can directly affect both the shape and size of the upper vocal tract and, consequently, the first and second formant frequency ranges. Indeed, the F1 /a/ is significantly altered (926 Hz vs. 1086 Hz between pediatric Crisponi and HS, *p* = 0.02, Table 2), suggesting that the frequent paroxysmal tightening of larynx muscles may have a direct detrimental effect on pharyngeal cavity resonance. Moreover, the formant ratios F1a/F1i (1.84 vs. 1.89 between Crisponi and HS, *p* = 0.04, Figure 1) and F1a/F1u (1.60 vs. 1.76 between Crisponi and HS, *p* = 0.04, Figure 1) were also found to be significantly lower in the pediatric pathological population. A difficulty in lowering the jaw does not allow the tongue to execute the appropriate vertical movements to reach the correct position to articulate vowels /i/ and /u/, which the ratios were able to detect. Additionally, as Figure 1 shows, the VSA is significantly lower than the HS group, further indicating the presence of articulation disorders, which was also confirmed by the perceptual analysis. For the AF group, besides NNE /i/ and NNE /u/, two other significant differences were discovered for the standard deviation of F1 /i/, including an instability that the laryngospasms could cause, and the F3 /i/, suggesting that articulation could also be altered at the uppermost part of the vocal tract, i.e., the lips. Indeed, the third formant is related to lip rounding capabilities [24] and the abnormal contraction of facial muscles that characterizes Crisponi syndrome [4], which may impede their correct positioning and shaping. Interestingly, FCR and VSA seem not to be altered in the adult population; this could be due to the smaller sample size. Nevertheless, more data, possibly coming from different vocal tasks such as standardized passages and the number listing tasks, are needed to validate this result. Notably, the most statistically significant differences concern the vocalization of /i/ and /u/. This result could suggest that when genetic syndromes are not characterized by striking voice properties, as occurs for the Cri du Chat syndrome [25], it is important to include in the recording protocol the utterance of additional vowel sounds that require more complex articulator mobility to better detect phonation and articulation deficits [10]. In conclusion, recent years have witnessed a significant expansion in the understanding of syndromes, neuromuscular disorders, and mitochondrial diseases. The extreme clinical variability of these conditions with multisystem involvement, the variable onset of symptoms from childhood to adulthood, and age-related comorbidities require the standardization of diagnostic and therapeutic coding [26]. The present research is a proof-of-principle study about the feasibility of undertaking a phonotype assessment in a relatively small cohort of patients with CS/CISS1. This study presents an opportunity for a more systematic approach to phenotypic characterization of voice in rare diseases. The phonotyping is of importance in the diagnostic work up of CS/CISS1 considering that this condition is underdiagnosed because the phenotype is relatively new and extremely complex. The implementation of phonotypes in the criteria for diagnosis of a rare and ultra-rare disease could be an example of how research on these diseases can be conducted with clinical spin-offs for patients and families.

## 5. Limits and Future Research

Our cohort’s age range is narrow, creating an opportunity for future research to comprehensively examine the clinical history of the syndrome in male adult patients. Furthermore, studies conducted in languages other than Italian could validate the findings and improve genotype—phenotype correlations associated with this rare genetic condition. Moreover, future studies on voice will also benefit from data obtained from laryngoscopy assessments. In the present paper researchers focused their work on the value of vocal phenotype in the diagnostic workup, rather than on evaluating the impact of voice or vocal changes on patients quality of life. This research, in conjunction with other previously conducted studies utilizing the same standardized methodology, will play a significant role in elucidating the human voice phenotype within the HPO. However, it may suffer from single assessment biases. In the future, with a greater sample size that also encompasses other diseases, artificial intelligence techniques could play a key role in studying the role of voice characterization in diagnostic work for genetic syndromes.

## Figures and Tables

**Figure 1 genes-16-00881-f001:**
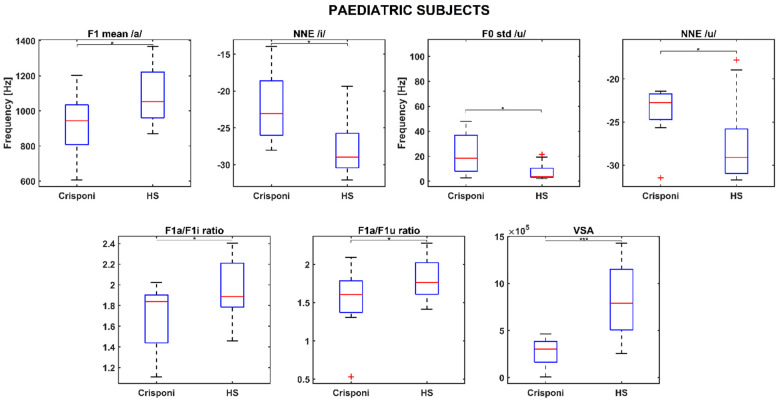
Boxplots for significant phonatory and articulatory parameters in the pediatric population. A (*) refers to a difference < 0.05, whereas (***) refers to a difference < 0.001. Statistical significance was estimated by a Mann–Whitney U test and reported accordingly: * *p* ≤ 0.05; *** *p* ≤ 0.001. HS = healthy subjects; NNE = normalized noise energy; VSA = vowel space area.

**Figure 2 genes-16-00881-f002:**
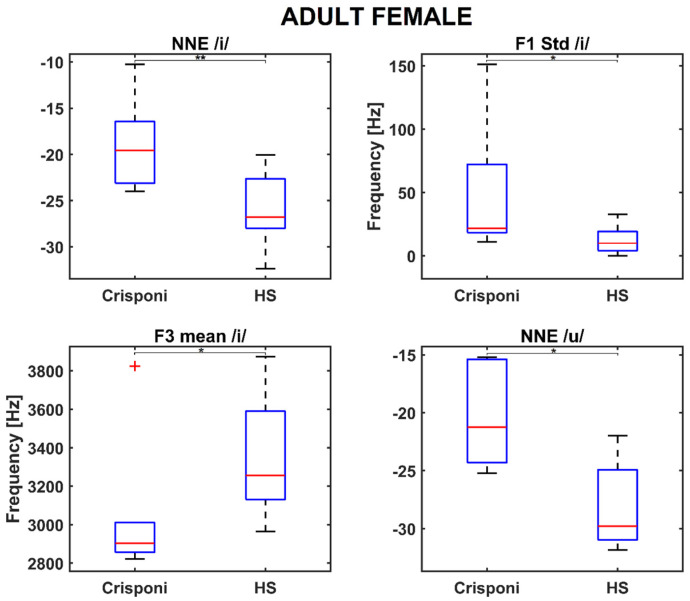
Boxplots for significant phonatory and articulatory parameters in the adult female population. A (*) refers to a difference < 0.05, whereas (**) refers to a difference < 0.01. Statistical significance was estimated by a Mann–Whitney U test and reported accordingly: * *p* ≤ 0.05. HS = healthy subjects; NNE = normalized noise energy.

**Table 1 genes-16-00881-t001:** GIRBAS (grade, instability, roughness, breathiness, asthenia, and strain) scale descriptive statistics presented as median (IQR).

Crisponi Subgroups	G	I	R	B	A	S
Group PS	1 (0.25)	0 (0)	0 (0)	0 (1)	0.5 (1)	0 (0)
Group AF	1 (0.5)	0 (0)	0 (0)	0 (0.5)	1 (0.5)	0 (0)
Group AM	1 (0)	1 (0)	1 (0)	0 (0)	1 (0)	0 (0)

Group PS, tot *n* = 7; group AF, tot *n* = 5; group AM, tot *n* = 1; G = grade; I = intensity; R = roughness; B = breathiness; A = asthenia; S = strain, PS = pediatric subjects; AF = adult females; AM = adult males.

**Table 2 genes-16-00881-t002:** GIRBAS (grade, instability, roughness, breathiness, asthenia, and strain) scale single evaluation of CS/CISS1 patients.

Patient ID #	Age (y)	Group	G	I	R	B	A	S	Speech Articulation Difficulties	Hypernasality
1	6	PS (F)	0	0	0	0	0	0	+	+
2	6	PS (M)	1	0	0	1	1	0	+	+
3	8	PS (M)	1	0	0	1	1	0	-	-
4	10	PS (F)	1	0	0	1	0	0	+	+
5	10	PS (M)	1	0	0	0	1	0	+	+
6	10	PS (M)	0	0	0	0	0	0	+	-
7	13	PS (F)	1	0	0	0	1	0	+	-
8	16	AF	0	0	0	0	0	0	+	+
9 *	17	AF	3	0	0	3	3	0	-	-
10	19	AF	1	0	0	1	1	0	+	+
11	44	AF	1	0	0	0	1	0	+	-
12	30	AF	1	0	0	0	1	0	-	+
13	25	AM	1	1	1	0	1	0	-	+

+ = presence; - = absence; y = years; PS = pediatric subject AF = female adult; AM = adult male; G = grade; I = intensity; R = roughness; B = breathiness; A = asthenia; S = strain. * Unilateral vocal cord paralysis.

## Data Availability

The datasets generated during the current study are available from the corresponding author upon reasonable request.

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
