# Peer review of "Do Rare Genetic Conditions Exhibit a Specific Phonotype? A Comprehensive Description of the Vocal Traits Associated with Crisponi/Cold-Induced Sweating Syndrome Type 1"

_genes, 2025, doi:10.3390/genes16080881_

Round 1
Reviewer 1 Report
Comments and Suggestions for Authors
I read with interest the paper titled: ‘Do Rare Genetic Conditions Exhibit a Specific Phonotype? A Comprehensive Description of the Vocal Traits Associated with Crisponi/Cold-Induced Sweating Syndrome Type 1’. The authors have conducted a thorough study using a specific methodology with an age-matched control group. There appears to be a couple of phonotype characteristics that can be defined as human phenotype ontology terms. I was keen to find out exactly the specific phonotype in this rare condition, however I felt that the authors have not clearly explained the impact of their work in this field. I have given details which include some minor fixes and also some significant revisions below.
Abstract: lots of information about the age (lines 28-29), but not about age at onset, and no explanation of the abbreviation ‘GIRBAS’. Unclear about the impact of the finding – was the common characteristic seen at younger ages?
Introduction:
Unclear about why phonotype is important in characterizing rare genetic disorders. Need more background on how impactful the use of phonotype is. How does phonotype corroborate or align with other phenotypes in other syndromes, eg is there any element of diagnostic and classification prediction possible?
Also is the Crisponi syndrome under-recognised and do patients face a long diagnostic journey, even in Sardinia which has a relatively high prevalence for an ultra-rare disease?
Methods:
Why was this methodology picked and not another type of voice analysis eg by Vizza et al. (2019)?
Lines 92-97 – were the controls age and sex-matched to the cases? Were they also matched in terms of ethno-geographic location (to capture any effect of accent)?
Lines 107-108 – suggest that the acoustic analysis protocol is standardized by Frassineti et al, however the references (15,16) provided do not showcase any test of inter-rater validation and any statistics on how the protocol has been standardized across different ethnic groups / different accents etc.
Line 111: What is ENT and SLP?
Lines 114-117: why is it important to align to the Human Phenotype Ontology? What is already known? Are there very scarce HPO terms associated with this syndrome?
In the Objective Acoustic Analysis (line 118) section– how is there standardization across accents and dialects? Is there any effect of patient fatigue on the analyses?
Results:
Please expand on all abbreviations in Table titles and captions.
Line 151: How was inter-rater agreement determined? Is the 0.8 a proportion of same responses between the raters?
Table 3 should show a column which highlights the statistically different values, with p values, between cases and controls. The way the table is displayed is not easy to interpret. The table could be shifted into supplementary as I am not sure it adds much to Figure 1.
Line 182: What is NNE?
Figure 3: The correlation displayed doesn’t look convincing, there must be large confidence intervals due to the limited data points. Is this the best way to display the data? Is this correlation adding anything to clinical knowledge?
How do the analyses differ with age across groups?
Is there any association with the phonotype and other phenotypes recorded in the cohort eg feeding difficulties?
Discussion
The discussion of the benefit of the HPO at the end instead of a concluding paragraph, seems a bit out of place – so I think this should be re-written as the readership will be aware of the benefits of HPO.
It would be helpful to understand whether a speech and language therapist could have labelled the individuals in the cohort with the dysphonic phenotypes (using HPO) without the additional support of the tools in this study?
We need to get a better idea of how this type of additional understanding can impact patient care or disease understanding. Without a real link to the pathology in this condition or association with other phenotypes, it is difficult to understand the value of the findings described. I do appreciate this study would have taken a long time to conduct and I hope there will be some additional pieces of data to help inform the readership of the impact.
Author Response
Dear Editor and Reviewers,
We would like to thank you for the interest in our study. We have revised the manuscript as per comments, using the track changes mode to highlight the modifications.
We appreciate your valuable feedback and hope the changes meet your expectations.
Reviewer #1
I read with interest the paper titled: ‘Do Rare Genetic Conditions Exhibit a Specific Phonotype? A Comprehensive Description of the Vocal Traits Associated with Crisponi/Cold-Induced Sweating Syndrome Type 1’. The authors have conducted a thorough study using a specific methodology with an age-matched control group. There appears to be a couple of phonotype characteristics that can be defined as human phenotype ontology terms. I was keen to find out exactly the specific phonotype in this rare condition, however I felt that the authors have not clearly explained the impact of their work in this field. I have given details which include some minor fixes and also some significant revisions below.
Abstract: lots of information about the age (lines 28-29), but not about age at onset, and no explanation of the abbreviation ‘GIRBAS’. Unclear about the impact of the finding – was the common characteristic seen at younger ages?
AA: We thank the reviewer for the comment. Age at onset was not specified due to the congenital nature of the condition. We have therefore added the term ‘congenital’ in line #24. We explained the term ‘GIRBAS’.
Introduction:
Unclear about why phonotype is important in characterizing rare genetic disorders. Need more background on how impactful the use of phonotype is. How does phonotype corroborate or align with other phenotypes in other syndromes, eg is there any element of diagnostic and classification prediction possible? Also is the Crisponi syndrome under-recognised and do patients face a long diagnostic journey, even in Sardinia which has a relatively high prevalence for an ultra-rare disease?
AA: We thank the reviewer for the suggestion. In the introduction section, we added clarification on the importance of characterizing the phonotype of rare under-recognised genetic disorders.
Methods:
Why was this methodology picked and not another type of voice analysis eg by Vizza et al. (2019)?
AA: The study applied the methodology proposed in Calà et al. (2023) as it was specifically designed for conducting voice analysis in patients diagnosed with genetic syndromes. The protocol represents an adaption of the standardized SIFEL (Società Italiana di Foniatria e Logopededia) protocol to be usable also with the targeted population.
Lines 92-97 – were the controls age and sex-matched to the cases? Were they also matched in terms of ethno-geographic location (to capture any effect of accent)?
AA: The control group was age and sex-matched to the pathological population, except for the adult male subgroup, where only one individual was present. It was decided not to consider his acoustic features in statistical analysis, and they were simply reported in Table 3 for sake of completeness. The control group was not matched in terms of ethno-geographic location. However, the protocol comprised the phonation of the cardinal Italian vowels /a/, /i/ and /u/ which are known to be relatively independent from dialectal inflections, unlike /e/ and /o/.
Lines 107-108 – suggest that the acoustic analysis protocol is standardized by Frassineti et al, however the references (15,16) provided do not showcase any test of inter-rater validation and any statistics on how the protocol has been standardized across different ethnic groups / different accents etc.
AA: We thank the reviewer for the comment. We have removed the statement. As answered in a previous answer, the voice analysis protocol was taken from the work of Calà et al., and it is a revisitation of an already existing, well-established protocol in Italian clinical practice to be more easily applicable when working with patients diagnosed with genetic syndromes.
Line 111: What is ENT and SLP?
AA: We have removed abbreviations and included the full terms.
Lines 114-117: why is it important to align to the Human Phenotype Ontology? What is already known? Are there very scarce HPO terms associated with this syndrome?
AA: We added a specification in the method and discussion section about the importance of alignment with the HPO (it responds to an idea of standardize the description of voice phenotypes that ultimately facilitates accurate diagnosis and research, in the field of genetics and rare diseases) and what is already known on this syndrome.
In the Objective Acoustic Analysis (line 118) section– how is there standardization across accents and dialects? Is there any effect of patient fatigue on the analyses?
AA: Thank you for your comment. As previously answered, to account for possible accents only the cardinal vowels /a/, /i/ and /u/ were chosen for statistical analysis. This task is indeed relatively independent from that confounding factor. Additionally, the recording protocol is short, about 2 minute maximum circa, and fatigue was never reported by clinicians performing the acquisition.
Results:
Please expand on all abbreviations in Table titles and captions.
AA: We expanded all abbreviations in Table titles and captions.
Line 151: How was inter-rater agreement determined? Is the 0.8 a proportion of same responses between the raters?
AA: We specified in the method section that the strength of agreement among perceptual analysis assessors (inter-rater reliability) was calculated by Krippendorff’s α.
Table 3 should show a column which highlights the statistically different values, with p values, between cases and controls. The way the table is displayed is not easy to interpret. The table could be shifted into supplementary as I am not sure it adds much to Figure 1.
AA: Thank you for noticing this issue. We have highlighted in grey the acoustic parameters that differed between the pathological subgroup and its related control subgroup and added a graphical representation for statistical significance and strength. Notably, Table 3 was simply provided to show the mean values for the most common voice features and have a direct form of comparison between Crisponi and healthy subjects. In any case, we agree with the reviewer suggestion to move Table 3 in a supplementary material section.
Line 182: What is NNE?
AA: We specified that NNE refers to Normalized Noise Energy.
Figure 3: The correlation displayed doesn’t look convincing, there must be large confidence intervals due to the limited data points. Is this the best way to display the data? Is this correlation adding anything to clinical knowledge?
AA: We thank the reviewer for this comment. Indeed, data numerosity is limited and it was decided not to split the sample to avoid working with even less observation to perform the correlation analysis. Nevertheless, the relationships discovered in this study are unlikely to be applicable or comparable with other genetic disorders and there were large confidence intervals. Therefore, we decided to follow this reviewer’s suggestion and remove the analysis.
How do the analyses differ with age across groups?
AA: It was decided not to further split the sample for conducting the correlation analysis. IN any case, as per our previous response, we decided to remove the correlation analysis.
Is there any association with the phonotype and other phenotypes recorded in the cohort eg feeding difficulties?
AA: There were no significant associations between the vocal phenotype and other phenotypes recorded in the cohort.
Discussion
The discussion of the benefit of the HPO at the end instead of a concluding paragraph, seems a bit out of place – so I think this should be re-written as the readership will be aware of the benefits of HPO.
AA: We re-written the discussion section by inserting the benefit of the HPO.
It would be helpful to understand whether a speech and language therapist could have labelled the individuals in the cohort with the dysphonic phenotypes (using HPO) without the additional support of the tools in this study?
AA: In this study, in addition to the perceptual evaluation of the voice through the human ear (SLP), we decided to further analyse the voice using a tool (Biovoice), regardless of the results we might obtain, in order to remain consistent with the methodology already used in previous publications on the voice and rare syndromes.
We need to get a better idea of how this type of additional understanding can impact patient care or disease understanding. Without a real link to the pathology in this condition or association with other phenotypes, it is difficult to understand the value of the findings described. I do appreciate this study would have taken a long time to conduct and I hope there will be some additional pieces of data to help inform the readership of the impact.
AA: Our project starts from the consideration that some genetic abnormalities causing a specific recognizable phenotype could also determine a specific vocal phenotype, or rather a “phonotype”. Since vocal assessment is based on non-invasive and easily administered tests, vocal characterisation could be an informative tool in the diagnostic process and help both in defining the severity of clinical pictures and in performing genotype/phenotype correlations. Artificial intelligence techniques will play a key role in studying the role of voice characterisation in diagnostic work for genetic syndromes. In addition, speech analysis could support the evaluation of the effectiveness of speech therapy, drug treatment and other rehabilitation approaches.

Reviewer 2 Report
Comments and Suggestions for Authors
Dear Authors
The authors describe vocal phenotypes of patients with a rare congenital disorder, Crisponi/Cold-Induced Sweating Syndrome Type 1. This would be a novel and interesting approach for delineating this disorder and also could be a non-invasive method for detecting such an ultra-rare condition. Several issues have been found to be concerned as follows:
Do the authors consider that Crisponi/Cold-Induced Sweating Syndrome Type 1 is a spectrum of a single disorder caused by biallelic pathogenic variant in CRLF1? So the authors do not seem to distinguish these two conditions in this cohort, OK?
Did the variants in CRLF1 differ between patients or some were shared in several patients associated with a founder effect?
It would be clearer to show the subgroup (PS/AF/AM) in each patient (probably Patient 1-7: PS; 8-12: AF; 12: AM). Why did the authors the cut-off of pediatric/adult at the age of around 15 years? Also, is it statistically appropriate for this subgrouping because too small number of patient(s) were/was included in AF/AM compared with PS?
It would be kinder for readers to show how G, I, R, B, A, and S mean as well as their scores (e.g., 1, 0, o.5).
Were clinical phenotypes (speech articulation difficulties, hypernasality) relevant to the results of examinations?
Were fiberscopic data for vocal code (e.g., movement, thickness) available in this cohort, which might be useful in detecting some additional coincidental abnormalities not related to this condition?
Were such approaches applied on other genetic disorders and if so how was the result?
Author Response
Dear Editor and Reviewers,
We would like to thank you for the interest in our study. We have revised the manuscript as per comments, using the track changes mode to highlight the modifications.
We appreciate your valuable feedback and hope the changes meet your expectations.
Reviewer #2
Dear Authors
The authors describe vocal phenotypes of patients with a rare congenital disorder, Crisponi/Cold-Induced Sweating Syndrome Type 1. This would be a novel and interesting approach for delineating this disorder and also could be a non-invasive method for detecting such an ultra-rare condition. Several issues have been found to be concerned as follows:
Do the authors consider that Crisponi/Cold-Induced Sweating Syndrome Type 1 is a spectrum of a single disorder caused by biallelic pathogenic variant in CRLF1? So the authors do not seem to distinguish these two conditions in this cohort, OK?
AA: Crisponi/Cold-Induced Sweating Syndrome Type 1 is a spectrum within a single disorder caused by biallelic pathogenic variants in CRLF1. Therefore, it should be regarded as a single disease with varying severity, especially during the neonatal period. All study participants exhibited a severe phenotype.
Did the variants in CRLF1 differ between patients or some were shared in several patients associated with a founder effect?
AA: All study participants were of Italian descent, mainly Sardinian (10 out of 13, 77%). The variants differ among patients. Sardinian cases typically display two variants in CRLF1, either in a homozygous or compound heterozygous state, which were associated with a founder effect.
It would be clearer to show the subgroup (PS/AF/AM) in each patient (probably Patient 1-7: PS; 8-12: AF; 12: AM). Why did the authors the cut-off of pediatric/adult at the age of around 15 years? Also, is it statistically appropriate for this subgrouping because too small number of patient(s) were/was included in AF/AM compared with PS?
AA: We performed this age classification in line with previous papers produced on the subject. We also wanted to make our results comparable, in the future, with those already studied on other conditions. Also, in acoustic analysis, due to different sizes and configuration in vocal folds and vocal tract, it is necessary to divide the sample in age and gender groups to get a more proper overview of voice characteristics. We added in Table 1 the acronym to clarify the subgroup affiliation.
It would be kinder for readers to show how G, I, R, B, A, and S mean as well as their scores (e.g., 1, 0, o.5).
AA: Thank you for your comment. We added a brief explanation of what GIRBAS indices and their value in Table 1 mean.
Were clinical phenotypes (speech articulation difficulties, hypernasality) relevant to the results of examinations?
AA: In the abstract and in the discussion section, we specified the relevancy of these findings.
Were fiberscopic data for vocal code (e.g., movement, thickness) available in this cohort, which might be useful in detecting some additional coincidental abnormalities not related to this condition?
AA: We thank the reviewer for pointing this out. The fiberscopic data for vocal cord were not available in this cohort. We added this specification in the limit section of the paper.
Were such approaches applied on other genetic disorders and if so how was the result?
AA: In the discussion section, we added reference to other papers applying this approach in other genetic disorders (SMS, Costello, Noonan, Down and Cri du Chat syndromes). Results were promising, showing that genetic syndromes do present unique vocal features that not only distinguish them from healthy controls but also across genetic syndromes themselves.

Round 2
Reviewer 1 Report
Comments and Suggestions for Authors
I thank the authors for their response but still have some concerns about the paper and suggestions for improving the overall message within this paper so that it attracts a larger readership.
Introduction:
I am still not sure how voice changes impact these patients – for example, are these patients living long enough to have impacts on their social integration due to differences in voice? Any previous documented issues with voice – even anecdotally? The introduction also doesn’t explain why the research is of importance in this group – I wanted to know why/how understanding the phonotype can have a clinical impact? Could it help somehow with education strategies for the children?
Are there long diagnostic odysseys for these patients that the phonotype can improve the diagnostic yield by including the assessment in the workup of the individual?
There is no reference given for line 65-66- ‘Furthermore, in some evolutive conditions, voice features could represent a prognostic indicator.’
Methods:
Line 115-116: still no justification given for the protocol used versus other protocols.
In the methods – there has been no consideration of separately subgrouping the cases according to the HPO terms listed. Was it possible to accurately apply the HPO terms through the thorough voice phenotype assessment or were there still some variability/inter-rater variance in how the HPO terms could be applied vs the voice assessment outcomes?
Were there significant changes in voice that cannot be captured by the HPO but only by the voice assessment?
Results:
Also it should be clear if the healthy subjects are age-matched controls – especially for results displayed in figures 1 and 2. It might be more interesting to display the differences for the different parameters between paediatric and adult patients with the same condition – to show whether you can depict a natural history of the phonotype in this condition. You should discuss the parameters being assessed for significant differences in the statistical analysis section.
Table 2: there is no need for the Gene column as everyone has the gene variant.
Discussion:
This is a proof-of-principle study about the feasibility of undertaking a phonotype assessment in a small cohort of patients with CISS1. - This needs to be made clear
Is CS/CISS1 poorly recognised as an entity? Is this why the phonotype is of importance in the diagnostic work up? – this needs to be answered in the discussion.
How would you use this work to differentiate between CISS2 and SWS which are the main other differentials?
Isn’t the main limitation that this is a single assessment? How do we know that the same results would be obtained on repetition of the task?
What about the natural history of the phonotype in rare diseases? Could that be of interest – particularly if related to any other negative outcomes for individuals with rare diseases affecting the voice?
There needs to be more discussion and referencing of papers that already consider AI technologies with voice recordings.
Minor:
Please check grammar throughout, here are some inconsistencies:
Line 54-54: Feeding difficulties are most present at birth, also caused by facial muscle contractions and orofacial weakness [7]. – this doesn’t make grammatical sense.
Lines 87-90, don’t need ‘n =’ Just say ‘6 paediatric patients…’
Line 158: cal-culated by Krippendorff’s α.
Line 233: phono type
Formatting of lines 240-248 needs to be looked at
Lines 334-343: repeats previous section
Author Response
Dear Editor and Reviewers,
we are submitting the revised version of our manuscript
We hope that the manuscript in its present form is suitable for publication.
We confirm that none of the original material contained in the manuscript has been submitted elsewhere.
All contributors approved the submission of this revised version of the paper for the journal.
Reviewer #1
I thank the authors for their response but still have some concerns about the paper and suggestions for improving the overall message within this paper so that it attracts a larger readership.
Introduction:
I am still not sure how voice changes impact these patients – for example, are these patients living long enough to have impacts on their social integration due to differences in voice? Any previous documented issues with voice – even anecdotally?
The introduction also doesn’t explain why the research is of importance in this group – I wanted to know why/how understanding the phonotype can have a clinical impact? Could it help somehow with education strategies for the children?
A: In the Limits and Future Research section we have now specified that in the present paper researchers focused their work on the on the value of vocal phonotype in the diagnostic field , rather than on evaluating the impact of voice or vocal changes on patients (e.g.social integration).
We believe that the definition of the vocal phonotype is important in rare conditions because this can be an additional diagnostic element , especially in those conditions where the clinical phenotype is mild.
In the specifics of this work, despite the extreme rarity of this specific condition, we were able to enroll as many as 13 patients, considering that isolated cases are often reported in the literature.
We thank the reviewer for the suggestion regarding the evaluation of the impact of vocal characteristics on patients' QoL, a topic to be integrated into future research.
Are there long diagnostic odysseys for these patients that the phonotype can improve the diagnostic yield by including the assessment in the workup of the individual?
A: Acoustic analysis is a fast, inexpensive, and noninvasive investigative technique that, if properly implemented can improve the diagnostic yield and could provide support for the description and characterization of genetic syndromes. The work presented lays the foundation for effective acoustic characterization of genetic syndromes.
There is no reference given for line 65-66- ‘Furthermore, in some evolutive conditions, voice features could represent a prognostic indicator.’
A: Lines 65-65 have been deleted.
Methods:
Line 115-116: still no justification given for the protocol used versus other protocols.
A: we thank the Reviewer for the comment. A clarification on the specific protocol usage is now included in the manuscript body, which reads "The choice of the vocal task, as well as the feature extraction pipeline explained in Section 2.4, followed the recommendations introduced in [10]. The latter represents an adapted version of the SIFEL (Società Italiana di Foniatria e Logopedia) protocol that was designed to be used with genetic syndromes patients, as it accounts for their voice production and behavioural impairments."
In the methods – there has been no consideration of separately subgrouping the cases according to the HPO terms listed. Was it possible to accurately apply the HPO terms through the thorough voice phenotype assessment or were there still some variability/inter-rater variance in how the HPO terms could be applied vs the voice assessment outcomes?
Were there significant changes in voice that cannot be captured by the HPO but only by the voice assessment?
A: In addition to the application of the GIRBAS scale, we have included in the perceptual analysis the term HPOhypernasality (identifier: 0001611) as it was already reported for this condition and the term articulation difficulties (HPO identifier: 0009088) as it was considered valuable during voices recordings. Acoustical parameters e.g. F1/F2/F3 etc. were calculated numerically rather that through qualitative description by HPO, and numerical results were displayed in the result section.
Results:
Also it should be clear if the healthy subjects are age-matched controls – especially for results displayed in figures 1 and 2. It might be more interesting to display the differences for the different parameters between paediatric and adult patients with the same condition – to show whether you can depict a natural history of the phonotype in this condition. You should discuss the parameters being assessed for significant differences in the statistical analysis section.
A: the Reviewer is right. We have now specified in the Methods that healthy control group is age and gender-matched. To improve clarity, it was decided not to produce figures (e.g., boxplots or similar) regarding the same pathological population to avoid readers thinking that statistical comparisons were made between paediatric and adult female groups. However, the possible evolution of acoustic properties can be deduced from Supplementary Table 1. Finally, we thank the reviewer for noticing this lacking detail in the Statistical Analysis section: we have now clarified that statistical analysis concerned only the objective parameters extracted and derived with the BioVoice toolbox.
Table 2: there is no need for the Gene column as everyone has the gene variant.
A: Gene column have been deleted.
Discussion:
This is a proof-of-principle study about the feasibility of undertaking a phonotype assessment in a small cohort of patients with CISS1. - This needs to be made clear
A: A specificication have been added.
Is CS/CISS1 poorly recognised as an entity? Is this why the phonotype is of importance in the diagnostic work up? – this needs to be answered in the discussion.
A: We thank the reviewer for pointing this out.A specificication have been added.
How would you use this work to differentiate between CISS2 and SWS which are the main other differentials?
A: As specifcied in the limit and future research section, other studies following the same standardized method are expected to serve this purpose.
Isn’t the main limitation that this is a single assessment? How do we know that the same results would be obtained on repetition of the task?
A: The reviewer is right. This limitation has now been specified in the Discussion.
What about the natural history of the phonotype in rare diseases? Could that be of interest – particularly if related to any other negative outcomes for individuals with rare diseases affecting the voice?
A: As specifcied in the limit and future research section, other studies following the same standardized method are expected to serve this purpose.
There needs to be more discussion and referencing of papers that already consider AI technologies with voice recordings.
A: We thank the reviewer for the suggestion. We are aware about the several studies that applied AI to speech pathology detection. However, this was out of the scope for this work (due to the low sample size), because it wanted to preliminarily point out potential differences in voice production. The application of AI has been simply suggested for future development but it might be not applicable in this voice-related research branch due to the extreme rarity of syndromes and language barriers. Therefore, it would not be appropriate to expand its discussion at this stage of the study.
Minor:
Please check grammar throughout, here are some inconsistencies:
A: We modified all the sentences.
Line 54-54: Feeding difficulties are most present at birth, also caused by facial muscle contractions and orofacial weakness [7]. – this doesn’t make grammatical sense.
Lines 87-90, don’t need ‘n =’ Just say ‘6 paediatric patients…’
Line 158: cal-culated by Krippendorff’s α.
Line 233: phono type
Formatting of lines 240-248 needs to be looked at
Lines 334-343: repeats previous section
Reviewer 2 Report
Comments and Suggestions for Authors
Dear Authors
Thank you for responding carefully each point I proposed.
The manuscript has been revised according to these comments.
Author Response
Many thanks